# Risk of stroke in male and female patients with atrial fibrillation in a nationwide cohort

Peter Brønnum Nielsen [ID][1,2,6], Rasmus Froberg Brøndum [ID][3], Anne Krogh Nøhr[3], Thure Filskov Overvad [ID][4] & Gregory Y. H. Lip [ID][1,5,6] ✉

Female sex has been suggested as a risk modifier for stroke in patients with atrial fibrillation (AF) with comorbid prevalent stroke risk factors. Management has evolved over time towards a holistic approach that may have diminished any sex difference in AF-related stroke. In a nationwide cohort of AF patients free from oral anticoagulant treatment, we examine the time trends in stroke risk overall and in relation to risk differences between male and female patients. Here we show that among 158,982 patients with AF (median age 78 years (IQR: 71 to 85); 52% female) the 1-year thromboembolic risk was highest between 1997–2000 with a risk of 5.6% and lowest between 2013–2016 with a risk of 3.8%, declining over the last two decades. The excess stroke risk for female vs male patients has also been declining, with risk-score adjusted relative risk estimates suggesting limited sex-difference in recent years.

Female sex has been suggested as a risk modifier for stroke in patients with atrial fibrillation (AF) with comorbid prevalent stroke risk factors. Female sex also has implications for oral anticoagulant (OAC) treatment decision-making, as it affects the $CHA_2DS_2$-VASc score, on which international treatment guidelines are based[1–3].

Prior studies have highlighted that despite female patients having a higher risk of AF-related stroke compared with male patients when left without anticoagulation, female patients tend to be under-treated with OAC[4,5]. Indeed, being female adds to the absolute stroke risk in patients aged ≥65 years or in the presence of ≥1 additional non-sex stroke risk factors[6]. The inclusion of female sex as a risk modifier into the $CHA_2DS_2$-VASc score may have aided the increase in OAC use in female patients, such that the male-female differences in OAC prescribing for stroke prevention in recently published AF cohorts are minimal[7].

Nevertheless, overall stroke rates appear to be declining over recent years, and AF management has evolved over time towards a holistic approach that may have helped reduce any sex differences in AF-related stroke[8]. Although older cohorts and meta-analyses have consistently shown that female patients with AF are at higher risk of stroke compared to male patients, more contemporary data are lacking to justify if we can move away from considering female sex as a stroke risk modifier in AF in risk scores or guidelines[9–13]. Such evidence has the potential to impact future clinical decision-making regarding stroke prevention, prompting a consideration of a differentiated approach to stroke risk assessment in male and female patients.

The decision on whether to initiate anticoagulation relies on an estimate of a patient's risk if left untreated, and the $CHA_2DS_2$-VASc score remains the recommended risk score for this purpose. Therefore, the aim of nationwide cohort study was to examine the time trends in thromboembolic risk overall and in relation to risk differences in male and female patients with AF who did not receive OAC treatment. Second, we also examined sex differences in OAC prescribing patterns in patients with AF.

Here we show that among patients with AF, the 1-year thromboembolic risk was highest in the earliest years and declining over the last two decades, with only marginal risk differences between female and male patients in the most recent time period.

[1]Danish Center for Health Services Research, Department of Clinical Medicine, Aalborg University, Aalborg University Hospital, Aalborg, Denmark. [2]Department of Cardiology, Aalborg University Hospital, Aalborg, Denmark. [3]Center for Clinical Data Science, Aalborg University and Aalborg University Hospital, Aalborg, Denmark. [4]Department of Clinical Pharmacology, Aalborg University Hospital, Aalborg, Denmark. [5]Liverpool Centre for Cardiovascular Science at University of Liverpool, Liverpool John Moores University and Liverpool Heart & Chest Hospital, Liverpool, UK. [6]These authors contributed equally: Peter Brønnum Nielsen, Gregory Y. H. Lip. ✉e-mail: gregory.lip@liverpool.ac.uk

## Results

A total of 408,930 patients with AF were considered for inclusion between year 1997 and 2020. After applying exclusion criteria, the study population consisted of 158,982 patients with AF (median age 78 years (IQR: 70 to 85); 52% female) who were not anticoagulated 30 days after the date of diagnosis (see Fig. 1). The total number of patients diagnosed was highest in the period from year 2009–2012 ($N = 32,170$) and lowest in the most recent period (2017–2020, $N = 13,017$) (Supplementary Table 2). The median non-sex specific $CHA_2DS_2$-VA score was 3 (IQR: 2–4) and it was similar for both male and female patients. Hypertension was common (50.8%) and 20.0% had a prior stroke, and approximately 45% had during the preceding year claimed a prescription for antiplatelet therapy with either aspirin or clopidogrel at the time of AF diagnosis (see Table 1 and Supplementary Table 3 for treatment at baseline).

During the study period, a total of 7538 thromboembolic events occurred, where the majority was ischemic strokes (94%). The 1-year incidence rate of thromboembolic events was high: 5.48 (per 100 person-years) among male patients and 6.88 among female patients. Over the 23-year study period, we observed a steady decline in event rates for both male and female patients (see Fig. 2). For female patients, the event rate peaked in year 1999 (8.32) and with a nadir in year 2015 (5.25), while for male patients the event rate was highest in 2019 and lowest in 2013, with rates of 6.41 and 4.24, respectively.

### Risk of thromboembolism

The overall 1-year thromboembolic risk was highest between 1997–2000 with a risk of 5.6% and lowest between 2013–2016 with a risk of 3.8%, declining over the last two decades, with male-female risk differences being less evident in recent years. The 1-year thromboembolic risk during the entire study period was higher in female patients (4.2% in male and 5.2% in female patients) and increased with higher $CHA_2DS_2$-VA score levels (Supplementary Table 4). The formal test for an overall time trend was statistically non-significant ($p = 0.12$).

The relative risks contrasting female vs male patients adjusted for $CHA_2DS_2$-VA score levels for time periods are presented in Fig. 3. During the follow-up period, the overall male-female relative risk diminished from 1.16 (95%CI 1.05–1.29) in years 1997–2000 to being close to similar, 1.05 (95%CI 0.86–1.25), in years 2017–2020. The observed trend in decreasing relative risk over time was not significant in the test for time trends ($p = 0.66$).

The effect modification of sex on stroke risk by points on the $CHA_2DS_2$-VA score stratified by time period is displayed in Table 2. In the time period 1997–2000, the associated risk of thromboembolism was 1.36 times higher among female patients compared with male patients for a score level of 2. In the most recent time period between 2017–2020, the associated risk of thromboembolism was highest at a score level of 4 (1.13 (95% CI 0.75–1.51)) but did not display significantly higher risks among female vs male patients at any score levels.

### Table 1 | Baseline characteristics

| Characteristic, % (N) | Male | Female | All |
|---|---|---|---|
| N | 76268 | 82714 | 158982 |
| Year 1997–2000 | 17.9 (13663) | 19.2 (15869) | 18.6 (29532) |
| Year 2001–2004 | 19.1 (14592) | 20.5 (16951) | 19.8 (31543) |
| Year 2005–2008 | 18.3 (13964) | 19.5 (16143) | 18.9 (30107) |
| Year 2009–2012 | 20.1 (15368) | 20.3 (16802) | 20.2 (32170) |
| Year 2013–2016 | 15.2 (11628) | 13.3 (10985) | 14.2 (22613) |
| Year 2017–2020 | 9.2 (7053) | 7.2 (5964) | 8.2 (13017) |
| Primary AF diagnosis | 45.5 (34715) | 45.0 (37183) | 45.2 (71898) |
| Age, median (IQR) | 75.0 (68.0–83.0) | 81.0 (73.0–87.0) | 78.0 (70.0–85.0) |
| $CHA_2DS_2$-VA, median (IQR) | 3.0 (2.0–4.0) | 3.0 (2.0–4.0) | 3.0 (2.0–4.0) |
| HASBLED, median (IQR) | 2.0 (1.0–3.0) | 2.0 (2.0–3.0) | 2.0 (2.0–3.0) |
| Heart failure | 27.1 (20631) | 26.5 (21905) | 26.8 (42536) |
| Hypertension | 49.4 (37683) | 52.1 (43089) | 50.8 (80772) |
| Diabetes | 16.6 (12698) | 13.0 (10724) | 14.7 (23422) |
| Prior stroke | 20.1 (15363) | 19.9 (16500) | 20.0 (31863) |
| Vascular disease | 22.0 (16810) | 15.8 (13065) | 18.8 (29875) |
| Prior bleeding | 15.6 (11911) | 14.2 (11751) | 14.9 (23662) |
| Myocardial infarction | 16.4 (12492) | 10.8 (8917) | 13.5 (21409) |
| Ischemic heart disease | 33.7 (25710) | 26.1 (21561) | 29.7 (47271) |
| Hyperlipidemia | 12.9 (9865) | 8.8 (7256) | 10.8 (17121) |
| PCI | 7.0 (5341) | 3.0 (2520) | 4.9 (7861) |
| CABG | 4.6 (3545) | 1.3 (1096) | 2.9 (4641) |
| COPD | 14.2 (10811) | 13.2 (10938) | 13.7 (21749) |
| Alcohol | 6.1 (4653) | 2.3 (1882) | 4.1 (6535) |
| Cancer | 20.7 (15776) | 17.9 (14834) | 19.3 (30610) |

*IQR* Interquartile range, *PCI* Percutaneous cardiac intervention, *CABG* Coronary artery bypass grafting, *COPD* Chronic obstructive pulmonary disease.

### Additional and sensitivity analyses

Female patients were less likely to start OAC in the early period (1997–2000), but OAC initiation increased over time and was highest in most recent time period (2017–2020), see Fig. 4. The pattern for OAC treatment initiation was similar for male and female patients, but consistently lower for female patients throughout the study period. A substantial increase in proportion of OAC treatment initiation was observed from the period of 2009–12 to 2013–16: the increase in proportion among male patients was from 18.0% to 30.4%, and from 14.5% to 26.1% among female patients.

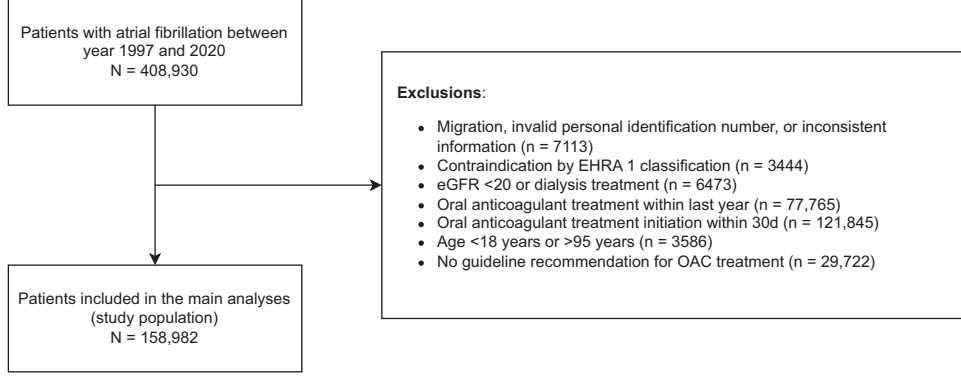

**Fig. 1 | Flowchart of the study population.** Flowchart shows the patients includes and excluded from our cohort.

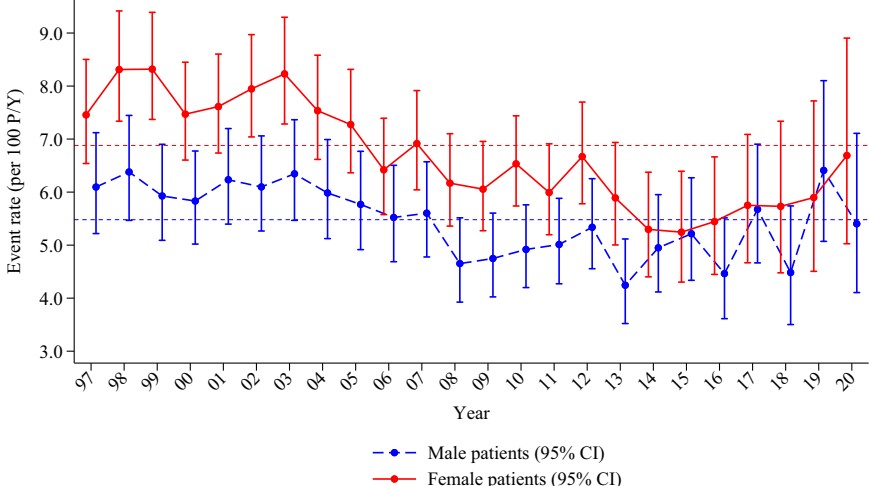

**Fig. 2 | Event rates.** Crude annual event rates of thromboembolism in atrial fibrillation per 100 person-years stratified by sex. Horizontal lines indicate the population mean event rates per 100 person-years throughout the study period (female 6.88 and male 5.48) Vertical bars represent confidence intervals (CI). See Source Data File for specific data points.

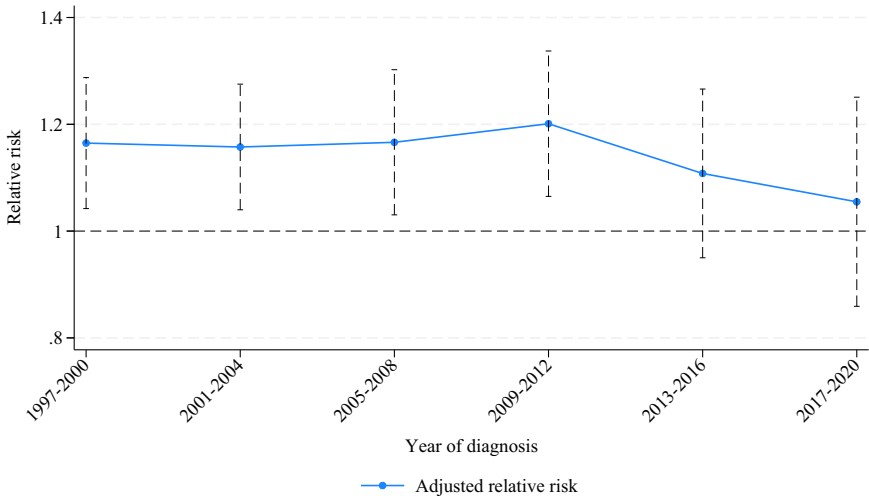

**Fig. 3 | Relative risk for male vs female patients.** Relative risk (and 95% confidence intervals as vertical bars) of thromboembolism contrasting male (reference) vs female risk according to study period and adjusted for points on the (non-sex specific) CHA2DS2-VA score. See Source Data File for specific data points.

**Table 2 | Relative risk of stroke for female vs male patients with newly diagnosed atrial fibrillation according to CHA$_2$DS$_2$-VA score levels in each study period**

|           | Score 1          | Score 2          | Score 3          | Score 4          | Score 5+         |
|-----------|------------------|------------------|------------------|------------------|------------------|
| 1997–2000 | 1.26 (0.73–1.80) | 1.36 (1.05–1.70) | 1.35 (1.05–1.65) | 1.01 (0.84–1.19) | 1.20 (0.99–1.42) |
| 2001–2004 | 0.98 (0.48–1.47) | 1.39 (1.07–1.71) | 1.07 (0.85–1.28) | 1.12 (0.91–1.33) | 1.16 (0.98–1.35) |
| 2005–2008 | 0.85 (0.44–1.26) | 1.09 (0.81–1.36) | 1.10 (0.84–1.35) | 1.16 (0.91–1.41) | 1.21 (1.00–1.42) |
| 2009–2012 | 2.17 (1.28–3.06) | 1.06 (0.77–1.34) | 1.23 (0.94–1.53) | 1.04 (0.82–1.28) | 1.28 (1.06–1.50) |
| 2013–2016 | 1.42 (0.50–2.35) | 0.97 (0.62–1.32) | 1.01 (0.71–1.31) | 0.91 (0.67–1.14) | 1.24 (0.99–1.50) |
| 2017–2020 | 0.76 (0.16–1.38) | 0.98 (0.60–1.36) | 1.08 (0.67–1.49) | 1.13 (0.75–1.51) | 1.01 (0.73–1.29) |

OAC sensitivity analyses by censoring at-risk time when initiation treatment found no meaningful difference and were consistent with our overall findings with a relative risk estimate between male and female patients of 1.18 (95%CI 1.07–1.31) in the years from 1997–2000 and 1.09 (95%CI 0.91–1.32) in most recent years. Confining the population to patients with AF and no records of cancer diagnosis within the last five years did not affect the observations on similarly stroke risk at most recent years: the relative risk between male and female patients

was 1.19 (95%CI 1.07–1.32) in years 1997–2000 and 0.93 (95%CI 0.77–1.12), in years 2017–2020. Supplementary Fig. 1 shows the 1-year risk of all-cause mortality stratified by sex, which was close to 30% for both male and female patients.

## Discussion

In this nationwide cohort study, our principal findings on patients with incident AF diagnosis who were not in OAC treatment are as follows: (i)

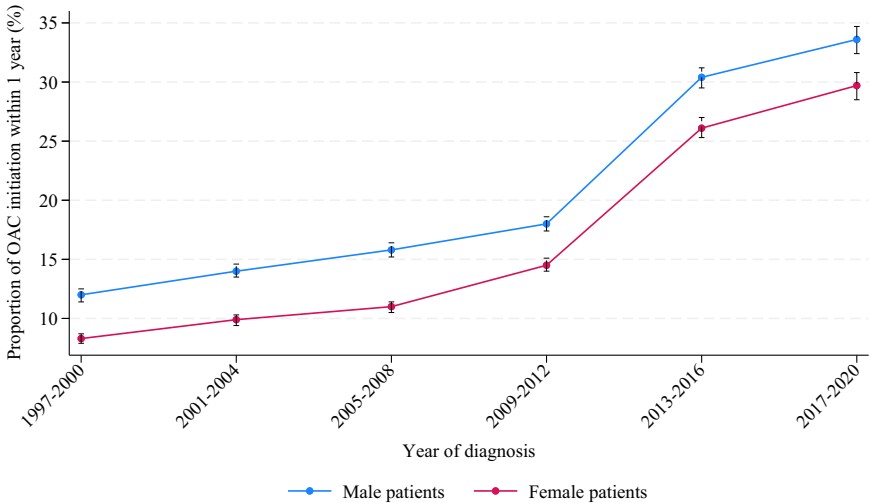

**Fig. 4 | Initiation of oral anticoagulant treatment.** Proportion of patients initiating OAC treatment within first year after atrial fibrillation diagnosis (vertical bars represent 95% confidence intervals). See Source Data File for specific data points.

the risk of stroke overall among patients with AF has been declining over the last two decades; (ii) the excess stroke risk observed for female vs male patients has correspondingly been declining, with relative risk estimates suggesting limited contemporary differences; and (iii) female patients were less likely to start OAC in all time periods, but OAC initiation increased over time and was highest in latest time period, and with similar male-female OAC initiation patterns.

Population studies have shown a decline in the overall strokes and AF-related stroke over the last decades. In the Swedish population, the rate of all ischaemic strokes steadily declined over the period 2001–2020[14]. By the end of 2020, 24% of all ischemic strokes still had a preceding or concurrent AF diagnosis, slightly higher than observed in 2001. Thus, even though both the absolute and relative risk of AF-related ischemic stroke declined over the past 20 years, every fourth ischemic stroke in 2020 still had a preceding or concurrent AF diagnosis[14].

In the present study, the decline in AF-related strokes was also clear, although whether this decline was due to differences in population characteristics, greater stroke risk factor management, and/or earlier detection and treatment of AF patients merits further study. Indeed, a more holistic or integrated care approach to AF care has been advocated, beyond anticoagulation for stroke prevention per se, with additional patient-centred decisions on rate or rhythm control, as well as proactive lifestyle and comorbidity management, summed up as the Atrial fibrillation Better Care (ABC) pathway[8]. The latter evidence-based integrated care management approach is now recommended in guidelines[1,3], given the profound benefits on mortality, stroke, and bleeding by adherence to the ABC pathway[15]. Our observations of an overall declining stroke risk were likely affected by this shift in management and could explain the substantial lower inclusion into the cohort in more recent years (see Supplementary Table 2). Indeed, in the 4-year periods from 2012 and onwards, the cohort inclusion dropped by approximately 1/3 in comparison with the previous time period. While a general shift in AF demographics and diagnosis was not explored, the exclusion of patients based on OAC treatment initiation increased during the same time period. Nonetheless, all-cause mortality risk was substantial in our cohort and close to 30% at 1 year follow-up. A previous study from the Framingham Heart Study based on data dating back 45 years (most recent inclusion year being 2015) suggested no evidence of temporal trends for all-cause mortality among patients with AF[16].

Female sex has been reported to be associated with a higher risk of AF-related stroke. In the present study, we confirm these observations of female patients having higher overall stroke rates compared with male patients, but the relative sex differences became smaller over time, across the spectrum of risk score levels, with the relative risk difference becoming non-significantly different in later years.

In the Stroke in AF Working Group systematic review of seven studies, female sex was an independent significant predictor of stroke in three studies (range of individual relative risks 1.6 to 1.9)[17]. A prior systematic review by Wagstaff et al. of 17 studies revealed a 1.30 (95% CI 1.18 to 1.46) elevated risk of stroke in women with AF[13]. A more recent systematic review and meta-analysis of cohort studies investigating male-female differences in clinical outcomes found a higher relative risk for stroke in female compared with male patients (RR 1.99, 1.46 to 2.71)[9]. Indeed, these data suggested that AF was associated with a greater relative risk of all-cause mortality, cardiovascular mortality, strokes, coronary heart disease, and heart failure in female than male patients.

In the ORBIT-AF registry[10], female patients with AF had an adjusted hazard rate ratio of 1.39 (1.05 to 1.84) for an excess risk of stroke and systemic thromboembolism compared with male patients, while in the Swedish nationwide cohort study[11], the annual stroke rate tended to be higher in female than in male patients with lone AF (age <65 years and no vascular disease), although the difference was not statistically significant (0.7% versus 0.5%, p = 0.09). Apart from these lone AF patients, women with AF had a modestly increase adjusted risk of stroke compared with men, suggesting that female sex should still be considered when making decisions about anticoagulation treatment. In a Canadian cohort study, Tsadok et al. found that female sex had an adjusted HR 1.14 (95%CI 1.07 to 1.22) for an increased risk of stroke events[18].

However, many of these prior AF cohorts included only older patients and/or those admitted with a recent diagnosis of AF. Also, an age-dependency to stroke risk was evident, as well as an additive risk of stroke in the presence of ≥1 stroke risk factors. In the Danish nationwide registries, for example, Nielsen et al. and others[6,19] reported that while the adjusted risk ratio for stroke was also higher in female AF patients compared with male patients, the risk of thromboembolism across score points only showed an excess of stroke rates in the presence of ≥1 additional stroke risk factors amongst female compared with male patients. Other factors such as inequities in cardiovascular care may partly explain higher stroke rates in female patients with AF. This perspective was confirmed in a different cohort from Canada suggesting that inequities in cardiovascular care including cardiologist

assessments and statin treatment could be explanatory factors for higher stroke rates among female patients[20]. Given these observations, female sex has been considered a risk modifier, rather than a stroke risk factor per se in recent guidelines[1,3].

In the Tasmanian Atrial Fibrillation Study[21], female patients with AF were undertreated with OAC, irrespective of $CHA_2DS_2$-VA scores of 0, 1 or ≥2, and suggested that the introduction of $CHA_2DS_2$-VA stroke risk stratification by the Australian AF guidelines[22] could potentially contribute to under recognition of female sex as a stroke risk factor. In an Italian cohort, including patients diagnosed between 2002 and 2013, there was undertreatment of female patients with OAC, especially in recent years[23]. Over the long term follow up, female patients had a higher risk of stroke but a lower risk of major bleeding and all cause death. Notwithstanding their undertreatment with OAC, when female patients present with a stroke, these tended to be more severe strokes as measured by the National Institutes of Health Stroke Scale[24].

When considering European patients, Corica et al reviewed crude OAC prescribing data from the EuroHeart Survey, the EORP-AF pilot registry, EORP-AF long term registry and the GLORIA AF European cohorts reflecting time trends from year 2005 and onwards with regard to prescribing habits in terms of OAC[5]. This showed undertreatment of female patients in earlier cohorts, but with the sex differences declining in more recent cohorts. With the EORP-AF registries and the GLORIA-AF registries, for example, the proportions prescribed OAC with one non-sex stroke risk factor amongst female patients was >80%, with prescriptions even higher amongst female (88.5%) compared with male patients (83.6%) in the GLORIA-AF registry. The present study is therefore consistent with these observations, showing that female patients were less likely to start OAC in the early time periods, but OAC initiation increased over time and was highest in most recent time period, with broadly similar OAC initiation patterns.

Whether these OAC prescribing trends reflect guideline changes, which introduced female sex as a risk modifier in the $CHA_2DS_2$-VASc score, as well as the recommendations that patients will benefit from stroke prevention treatment unless the patient is at very low risk of stroke ($CHA_2DS_2$-VASc score 0 in male, or 1 in female patients) remains uncertain. However, given that the sex differences in stroke have become less evident in recent years, a practical consideration is whether female sex (Sc) criterion is still needed in the $CHA_2DS_2$-VASc score, when OAC initiation is needed, as we previously proposed in 2018[6,25] In our prior study, Nielsen et al. concluded that "female sex is a risk modifier for stroke in patients with AF and initial decisions on OAC treatment could be guided by a $CHA_2DS_2$-VA score". Nonetheless, we should continue to acknowledge that the Sc risk component modifies and accentuates stroke risk in women who would have been eligible for OAC treatment based on ≥2 additional stroke risk factors.

However, the available data on the $CHA_2DS_2$-VA score were limited and the published studies had methodological issues, including the largest validation study which was subsequently retracted for methodological reasons[26]. Also, time trends in sex differences in stroke rates suggest some decline, but only more recently data showed that male-female differences in AF-related stroke were increasingly minimal.

Given the limitations of all clinical risk scores, including their modest predictive value and the dynamic nature of stroke risk[27], as well as individual risk factors not carrying equal weight for stroke risk, more recent guidelines have moved towards a more simple and practical recommendation that the default for AF patients should be to offer stroke prevention unless the patient is at low risk. Indeed, the artificial risk categorisation in low, moderate, and high-risk strata, which has been used in older guidelines still resulted in suboptimal OAC prescribing in AF populations[28]. Hence, rather than focus on risk strata, decision-making for OAC should be more focused on a holistic approach towards risk factor management rather than artificial stroke risk categories.

This study is limited by its dependence of administrative datasets, where potentially important clinical information was not available to further characterise stroke risk or subtypes of stroke. There was a risk for misclassification of points assigned on the $CHA_2DS_2$-VA score, specifically to underestimation of the accurate score level if certain comorbidities are not captured by our register-based definitions, because the components included for calculating the score were based on hospital records and prescribed medication alone. Yet the potential misclassification is unlikely to be sex dependent but could theoretically contribute to an overestimation of risk for specific risk score levels that would characterize the cohort to have a lower stroke risk. The nationwide registries allowed for virtually complete follow-up, minimizing selection bias from differential loss to follow-up. Nevertheless, the decision to exclude patients in oral anticoagulant treatment had implications for our study findings, as the generalizability is limited to patients with AF who are not receiving stroke prevention treatment. As such, selection bias belonging to restriction of the study population may have implications resulting in higher stroke rates than what would have been observed if a mixed cohort of treated and untreated patients were included.

The precision of risk estimates was limited by the number of outcomes in subgroups of patients at some score levels. Data on the occurrence of thromboembolism were derived from a nationwide stroke registry with obligatory registration but could be underreported if patients died to stroke, since post-mortem is not mandatory in Denmark. Importantly, our study did not try to elucidate the biological effect of sex on risk of stroke in AF, but merely to describe the clinical history of sex differences in stroke risk over time both overall and in important subgroups, i.e., $CHA_2DS_2$-VA score levels, in order to evaluate whether future risk stratification strategies should continue to include sex as a factor important for OAC treatment decisions.

Our results reflect only time trends in sex differences in Denmark and conclusions may be confined to Caucasians only. Since clinical management may differ from country to country, time trends in other countries remains worthy of exploration.

In our study of patients with AF who were not using OAC treatment, the risk of stroke has been declining over the last two decades. The excess stroke risk for female vs male patients has also been declining, with risk-score adjusted relative risk estimates suggesting limited sex-difference in recent years. Stroke prevention with OAC has increased over time and was highest in the most recent time period, and whilst the initiation pattern of OAC was similar for both male and female patients, it was consistently lower for female patients.

## Methods

The study was conducted in compliance with Danish Data Protection Agency under institutional registration (original project number 2017–40) granting access to healthcare registry data. Ethical approval is not required according to Danish law when using deidentified data linked from healthcare registries; similarly, written informed consent was not required when analysing deidentified existing healthcare data for research purposes. This was an observational cohort study including patients with a hospital diagnosis of AF between 1997 and 2021. Data was obtained from the Danish nationwide registries through institutional registration and authorization conditions of access through the Danish Data Protection Agency and the Danish Health Data Authority and conducted in compliance with the General Data Protection.

The Danish National Patient Registry was used to identify patients with an AF diagnosis according to the International Classification of

Diseases (ICD) 10th revision (ICD10: I48) between January 1997 and December 2020[29]. Information from the Danish National Prescription Registry and the Danish Civil Registration System was cross-linked to obtain individual level information on sex, vital status, comorbidity, and medication status[30,31]. Last, information from the Danish National Laboratory Registry was included to identify patients with impaired renal function, which may contraindicate the use of oral anticoagulants for stroke prevention[32].

## Study population

All patients with an AF diagnosis registered in a Danish hospital, including both emergency room, inpatient and outpatients settings in the time period were identified. Patients handled solely in general practice were not included as these patients were unavailable in the data materials. Potentially eligible patients age ≤95 years were screened for exclusion criteria including records of valvular heart disease according to the EHRA classification (EHRA class 1)[33], and estimated glomerular filtration rate (eGFR) < 20 or records of dialysis (see Fig. 1). As we were interested in stroke risk among patients with AF who were not in oral anticoagulant treatment, we excluded patients who had claimed a prescription of vitamin K-antagonists, Factor Xa inhibitors, or Factor IIa inhibitors in the preceding year to AF diagnosis, or those who claimed a prescription within 30 days after the diagnosis. While this epidemiological approach may risk imposing a selection bias pertaining to the generalizability of the study results, we sought to study the intrinsic risk of stroke unconfounded by the use of anticoagulant treatment.

## Variables of interest

We abstracted clinical and demographic variables at the time of the AF diagnosis. Similarly, information on medical treatment was collected according to records in the Danish National Prescription Registry holding information on type of medication based on Anatomical Therapeutical Classification Chemical (ATC) system codes and the date of purchase. The $CHA_2DS_2$-VA score (excluding the Sc "Sex category" from the original score) was calculated using information on comorbidities from the complete hospital records history and concomitant medication based on the year before (see Supplementary Table 1). The applied registry information has previously been verified as suitable for epidemiological investigations with adequate validity[34]. Sex and gender both exist on a spectrum, and are not necessarily aligned. For simplicity, we chose to look only at male and female patients, as defined in the registry where this information was collected based on anatomy at birth.

## Outcome measures

The Danish National Patient Registry was utilized to track patients with the main outcome of 'thromboembolism', encompassing records of ischemic stroke and/or systemic embolism. Specifically, the outcome was determined from hospital records of patients with a primary diagnosis ICD10 code of I63, I64.9, or I74. To further explore the impact of potential reasons for changes in stroke rates, we investigated prescribing patterns using the first prescription claim of an OAC (ATC-codes: B01AE07; B01AF01; B01AF02; B01AF03; B01AA03; B01AA04) as an outcome. Last, all-cause mortality was included as a lone outcome to allow for inference in differential survival times in male and female patients.

## Statistical analyses

Survival analysis was applied to estimate the risk of thromboembolism in patients with AF categorized using baseline risk levels of the $CHA_2DS_2$-VA score and examined according to the date of the AF diagnosis. Score levels of 5 or greater were collapsed into one group as there were few patients above that level. At-risk time was measured from the date of AF diagnosis until the outcome of interest, emigration, death, end of follow-up (1 year after the index data), or end of study period (July 2021), whichever came first. A maximum of 1 year follow-up time was chosen, since thromboembolic risk is a continuum reflecting natural dynamic changes of increased risk with accumulation of additional stroke risk factors[35]. Crude event rates and unadjusted risk estimates were provided to describe the overall outcome data in the cohort.

To assess the association between sex and thromboembolism, we estimated the risk ratio for female vs male (reference) adjusted for points on the $CHA_2DS_2$-VA score. This approach was repeated for each of the following six time periods: 1997–2000, 2001–2004, 2005–2008, 2009–2012, 2013–2016, and 2017–2020. The relative risks were estimated by means of the pseudo-value approach taking into account the competing risk of death[36]. With the aim of exploring whether female sex serves as a risk modifier on the $CHA_2DS_2$-VA scale, we employed a generalized linear model with a log-link function on the event status, introducing sex as an interaction term on the $CHA_2DS_2$-VA score to estimate a potential effect modification by sex on the risk at each score levels. To formally test whether the risk profiles associated with $CHA_2DS_2$-VA score levels differed for men and women, we conducted a likelihood ratio test comparing the interaction model with the model having $CHA_2DS_2$-VA score and sex only as main effects. Finally, an interaction term for the full model was employed to test for a time trend by means of a log-likelihood ratio test for the model without the interaction term. Point estimates, along with 95% confidence intervals (CI), were presented, and statistical significance was considered at a P-value of <0.05. The analyses were conducted using STATA/MP (v. 18.0).

## Sensitivity and additional analyses

In the main analysis, patients with AF were non-anticoagulated at baseline. To investigate the proportion and impact of initiation of OAC on the risk of thromboembolism, the following analyses were conducted: First, a time-to-treatment initiation analysis was conducted using the first registered prescription claim as the outcome. The first prescription was found by searching the Danish National Prescription Registry for records of warfarin, phenprocoumon, apixaban, dabigatran, edoxaban, or rivaroxaban. Second, we censored follow-up time when a patient claimed a prescription of on OAC treatment. This was done to investigate the potential impact of mitigating stroke risk after OAC treatment initiation, while potentially imposing a risk of bias related to informative censoring. Last, we conducted a sensitivity analysis where patients with a cancer diagnosis within the last 5 years were excluded, since these patients may exhibit a different risk profile that potentially may affect the treatment decision of OAC treatment.

## Reporting summary

Further information on research design is available in the Nature Portfolio Reporting Summary linked to this article.

# Data availability

The data supporting the findings from this study are available within the manuscript and its supplementary information. Source data are provided for Figs. 2 to 4 in Supplemental Source Data File. Data protection rules prohibits individual level data to be shared. Permission to access can be obtained following approval from Danish Data Protection Agency and the Danish Health Data Authority. Updated details and conditions for access to the data can be found here: https://sundhedsdatastyrelsen.dk/da/english/health_data_and_registers/research_services/before_applying. Source data are provided with this paper.

## Code availability

Code specifically developed for data management and statistical analyses can be shared upon request.

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

## Author contributions

P.B.N. and G.Y.H.L. conceived and planned the study. P.B.N., A.K.N., and R.F.B. performed the technical parts and analytic approach. P.B.N., T.F.O., and G.Y.H.L. analysed the data and contributed to the original interpretation. P.B.N. wrote the draft manuscript and all authors discussed the results and contributed to the final manuscript.

## Competing interests

G.Y.H.L.: Consultant and speaker for BMS/Pfizer, Boehringer Ingelheim and Daiichi-Sankyo. No fees are received personally. P.B.N.: Consultant fees from Boehringer Ingelheim, grants and consultant fees from Daiichi-Sankoy, grants from BMS/Pfizer, and grants and consultant fees from Bayer outside the submitted work. Other authors: None.
