## [Peer Review File · Nature Communications]

REVIEWERS' COMMENTS

Reviewer #2 (Remarks to the Author):

The authors addressed all issues and improved the manuscript. The limitations of the study are adequately addressed

Reviewer #3 (Remarks to the Author):

This revised manuscript by Dr. Nielsen and colleagues has 2 aims: (1) examine the time trends in stroke risk overall and in relation to risk differences between males and females, and (2) examine sex differences in OAC prescribing patterns in patients with AF. The manuscript is improved in some respects after the revision, but a significant concern remains regarding selection bias.

The authors have focused on a sample of patients with AF who are not on oral anticoagulation (OAC) in order to meet their study aims. However, the same factors (many of which are unknown and therefore unmeasured) that have led certain patients not to be prescribed OACs (despite having indication for OAC) can very well also influence stroke and thromboembolic risk. It is interesting to note that about 20% of patients in this sample had history of stroke and yet were not on OAC. Why?

The only unbiased way to accurately assess intrinsic stroke risk is to enroll patients with AF who are not on OAC prospectively and follow them up over time with respect to stroke and thromboembolic events. This is obviously quite challenging to perform. Short of this, some approach to control for confounding by indication can potentially present findings that more accurately reflect the truth.

REVIEWERS' COMMENTS

Reviewer #2 (Remarks to the Author):

The authors addressed all issues and improved the manuscript. The limitations of the study are adequately addressed

>>> We thank the reviewer for the constructive contributions.

Reviewer #3 (Remarks to the Author):

This revised manuscript by Dr. Nielsen and colleagues has 2 aims: (1) examine the time trends in stroke risk overall and in relation to risk differences between males and females, and (2) examine sex differences in OAC prescribing patterns in patients with AF.

>>> We thank the reviewer for reading and commenting on our manuscript.

The manuscript is improved in some respects after the revision, but a significant concern remains regarding selection bias. The authors have focused on a sample of patients with AF who are not on oral anticoagulation (OAC) in order to meet their study aims. However, the same factors (many of which are unknown and therefore unmeasured) that have led certain patients not to be prescribed OACs (despite having indication for OAC) can very well also influence stroke and thromboembolic risk. It is interesting to note that about 20% of patients in this sample had history of stroke and yet were not on OAC. Why?

>>> We acknowledge the risk of selection bias that pertains to the generalizability of our sample to the AF cohort as a whole. However, as also mentioned by the Reviewer, we constructed the cohort to be fitting with the research question of interest. While out of scope of our current study, we post hoc explored a cohort confined to those with a diagnosis of prior stroke (20.0% of the original cohort). Here we noticed that 26.4% had a diagnosis of prior transient ischemic attack, and a total of 15% had a prior significant major bleeding. We were not able to identify other immediate related clinical factors, which may have impacted the decision on treatment.

The only unbiased way to accurately assess intrinsic stroke risk is to enroll patients with AF who are not on OAC prospectively and follow them up over time with respect to stroke and thromboembolic events. This is obviously quite challenging to perform. Short of this, some approach to control for confounding by indication can potentially present findings that more accurately reflect the truth.

>>> Thank you for this comment. We agree that inclusion of a cohort of patients with AF and without an indication for stroke prevention treatment could be prospectively followed to study intrinsic stroke risk. However, this approach would only answer the question of stroke risk among patients without an indication for anticoagulant treatment, as it would be unethical to enroll patients and not offer them state-of-the-art treatment as per guideline recommendations. Therefore, we respectfully maintain – within the limitations of the analytic approach – that our study design represents the most accurate way of studying stroke risk in a heterogeneous cohort of patients with AF without stroke prevention treatment.